# Object Detection in Autonomous Driving Scenarios Based on an Improved Faster-RCNN

**Yan Zhou [1],\*, Sijie Wen [1], Dongli Wang [1], Jinzhen Mu [2] and Irampaye Richard [3]**

[1]  School of Automation and Electronic Information, Xiangtan University, Xiangtan 411105, China; wensijie0711@163.com (S.W.); wangdl@xtu.edu.cn (D.W.)

[2]  Shanghai Aerospace Control Technology Institute, Shanghai 201109, China; jinzhen_mu@163.com

[3]  School of Mathematics and Computational Science, Xiangtan University, Xiangtan 411105, China; richarcive@gmail.com

\*  Correspondence: yanzhou@xtu.edu.cn

**Abstract:** Object detection is one of the key algorithms in automatic driving systems. Aiming at addressing the problem of false detection and the missed detection of both small and occluded objects in automatic driving scenarios, an improved Faster-RCNN object detection algorithm is proposed. First, deformable convolution and a spatial attention mechanism are used to improve the ResNet-50 backbone network to enhance the feature extraction of small objects; then, an improved feature pyramid structure is introduced to reduce the loss of features in the fusion process. Three cascade detectors are introduced to solve the problem of IOU (Intersection-Over-Union) threshold mismatch, and side-aware boundary localization is applied for frame regression. Finally, Soft-NMS (Soft Non-maximum Suppression) is used to remove bounding boxes to obtain the best results. The experimental results show that the improved Faster-RCNN can better detect small objects and occluded objects, and its accuracy is 7.7% and 4.1% respectively higher than that of the baseline in the eight categories selected from the COCO2017 and BDD100k data sets.

**Keywords:** autonomous driving; object detection; small objects; occluded objects; Faster-RCNN

## 1. Introduction

In recent years, with changes in market demand and the rapid development of the automotive industry, autonomous driving has become a research hotspot in the automotive field. At present, many Internet companies and automobile companies worldwide have set foot in the field of autonomous driving. Object detection is one of the key algorithms in automatic driving systems, so it is challenging to design an efficient object detection algorithm for this complex scenario.

In the field of autonomous driving, the main detected objects are divided into two categories: stationary objects and moving objects. Stationary objects include traffic signs, traffic lights, and obstacles, while moving objects include vehicles, pedestrians, and non-motorized vehicles. The detection of moving objects is particularly important, and there are several difficulties: (1) the object to be detected may have different degrees of occlusion; (2) the detection effect for small objects is not good; and (3) the requirements for detection speed and detection accuracy often cannot be met at the same time. Therefore, whether the above problems can be solved directly affects the safety performance of autonomous driving.

Traditional object detection algorithms mostly use sliding windows to extract features of different sizes and regions; then, corresponding classifiers are trained for specific objects, and the trained classifiers are used to classify the extracted features. Since traditional features are designed manually by humans, many of them are designed for specific object detection, and their use has limitations. In many cases, artificial design features cannot describe the essence of an image. Therefore, they are susceptible to various interference

factors during detection. The model is not robust and cannot be applied in autonomous driving scenarios.

The environment faced by autonomous driving is open and complex. To ensure driving safety, the performance requirements of detection algorithms are extremely high. Therefore, most of the current automatic driving schemes use object detection algorithms based on deep learning [1]. These algorithms can be divided into two categories: One type is the two-stage object detection algorithm based on candidate region classification, and the process can generally be divided into two steps. First, region proposals are extracted [2], and then the region proposals are classified and the position coordinates are corrected. This type of algorithm has high accuracy but slow speed. Such algorithms include the region-based convolutional neural network (R-CNN) [3], Fast-RCNN [4], Faster-RCNN [5], Mask-RCNN [6], Cascade-RCNN [7], Libra-RCNN [8], and various improved versions. The other type is the one-stage algorithm based on regression, which transforms the object detection problem into a classification problem. Examples of such algorithms are the single-shot detector (SSD) [9], you only look once (YOLO) [10], YOLO9000 [11], YOLOv3 [12], YOLOv4 [13], and various improved versions. This type of algorithm is fast and has a small model, but it has a very poor detection capability on small objects and is prone to missed and false detections (the $AP_s$ of yolov4 on the COCO dataset is 20.4%, and that of Faster-RCNN is 21.8%). In an autonomous driving system, it is necessary to detect distant objects to make decisions in advance. However, a distant object occupies only a few pixels in an image, which causes great difficulties for the one-stage object detection algorithm.

Taking into account the accuracy and detection rate of the model, we selected the Faster-RCNN algorithm as the basis and aimed to improve and optimize the Faster-RCNN algorithm for the problems of occlusion, denseness, and small scale in object detection tasks in autonomous driving scenarios. This paper thus proposes a new algorithm called the Automatic Drive-Faster-RCNN (AD-Faster-RCNN).

The main contributions of this paper are as follows: (1) Based on Faster-RCNN, a new model structure is proposed to improve the detection ability of small objects and occluded objects in autonomous driving scenarios. (2) On the basis of Resnet-50, deformable convolution [14] is used instead of the traditional convolution, and a partial attention mechanism [15] is added. AD-Resnet-50 (Attention Deformable-Resnet-50) with a stronger feature extraction ability is proposed. (3) Based on a feature pyramid network (FPN) [16], a new feature pyramid structure fused by a multiscale feature is proposed.

## 2. Related Work

### 2.1. Object Detection Algorithm Based on Deep Learning

At this stage, object detection is divided into one-stage and two-stage algorithms. The detectors with the highest detection accuracy at present are two-stage detectors. The most representative two-stage detector is R-CNN and its variants. R-CNN uses the selective search algorithm to extract suitable candidate proposals in the image, converts them to a standard size, and sends them to the convolutional network for feature extraction; finally, it uses a support vector machine (SVM) for classification. Girshick proposed Fast-RCNN to address the shortcomings of R-CNN, such as its slow detection speed and large amount of calculation it requires. Fast-RCNN performs softmax classification and candidate proposal regression on the feature vectors extracted from the region of interest (ROI), achieving a unified implementation of feature extraction and classification tasks. Ren et al. proposed the Faster-RCNN algorithm, which uses region proposal networks (RPNs) instead of selective search to recommend candidate proposals; this greatly improves the quality of candidate proposals, reduces the amount of calculation, and achieves end-to-end training classification. He et al. proposed Mask-RCNN and made two improvements based on previous studies, using ROI Align and Mask branches to achieve more accurate instance segmentation while reducing the running time. Zhao et al. proposed Cascade-RCNN to solve the problem of low sample quality for RPNs. Pang et al. proposed Libra-RCNN to

solve the three problems of sample imbalance, feature imbalance, and detection imbalance, which improved the detection efficiency.

A one-stage detector converts classification and positioning problems into regression problems and performs end-to-end detection [17]. The detection speed is greatly improved, but the accuracy is reduced. SSD incorporates the concept of anchors in Faster-RCNN, proposes a default box generation method, and uses different-scale feature maps to detect objects of different sizes, thereby improving the speed and accuracy of the algorithm. The YOLO series algorithm greatly improves the detection speed. YOLOv3 uses ResNet's [18] remaining block structure to solve the gradient dispersion and gradient explosion problems caused by the deepening of the network hierarchy; it also uses FPN [19] to fuse multiscale feature maps to improve detection accuracy.

### 2.2. Object Detection in Autonomous Driving Scenarios

Object detection algorithms based on deep learning are mainly used in vehicle detection, pedestrian detection, and traffic sign detection in autonomous driving scenarios. Wang et al. proposed a method that could effectively detect small vehicles in traffic scenes using multiscale feature fusion and focal loss [20]; Chen et al. used group convolution to design a lightweight detection network for autonomous driving scenarios [21]; Liu et al. proposed an efficient pedestrian detection algorithm based on SSD to solve the problem of missed detection in dense crowds [22]; Z Liu et al. proposed the deconvolution of a region-based convolutional neural network to solve the problem of small traffic sign detection [23].

Unlike previous work, this paper proposes an object detection algorithm based on Faster-RCNN in an autonomous driving scenario. The detected objects include vehicles, pedestrians, and common small objects such as traffic lights and stop signs. Through the improvement of each module of the original model, the previous shortcoming of poorly detecting small objects is solved, the detection performance of the model in automatic driving scenarios is improved, and the detection accuracy is improved while ensuring high speed.

## 3. Proposed Algorithm: AD-Faster-RCNN

### 3.1. Principle of AD-Faster-RCNN

Figure 1 shows the AD-Faster-RCNN model proposed in this paper. The backbone network uses Resnet-50 [18], which is mainly used for image feature information extraction tasks.

The backbone network of AD-Resnet-50 is composed of five large structural blocks (Res1–Res5). Res1 contains convolution, BN layer, ReLU activation function, and maximum pooling to reduce the image resolution. The role of the BN layer is to prevent the gradient from disappearing or exploding, and to speed up the training , while the role of the ReLU activation function is to increase the non-linearity of the network. Res2–Res5 are used to extract information from different dimensions. It is composed of the Conv Block and Identity Block, and adopts a residual connection structure so that layers can copy their inputs to the next layer.

In order to improve the model's adaptability to geometric deformation and the ability to extract small objects, we added a spatial attention mechanism to the last two structural blocks of the Resnet-50 backbone network; we also applied variability convolution V2 instead of the traditional convolution, and named it AD-Resnet-50. Then, the feature maps extracted from each layer of Res2–Res5 in the backbone network were fused using the improved feature pyramid (PAB-FPN) module to obtain feature maps Z2–Z6 of five scales, and then the fused feature maps were entered into the region proposal network to obtain candidate object regions. After the candidate region was selected, the ROI mapped to the original image was merged at a uniform size (7 × 7) through the ROI Align layer, and then two fully connected layers FC were entered for classification and regression. The candidate object area was resampled through the bounding box regression output by the previous detector, the IOU (Intersection-Over-Union) threshold was gradually increased, and finally,

a new classification score and bounding box regression were obtained by training. In the three cascade detectors, side-aware boundary localization was used to replace the traditional frame regression. Finally, the Soft-NMS (Soft Non-maximum Suppression) algorithm was used to remove the bounding boxes and retain the best results.

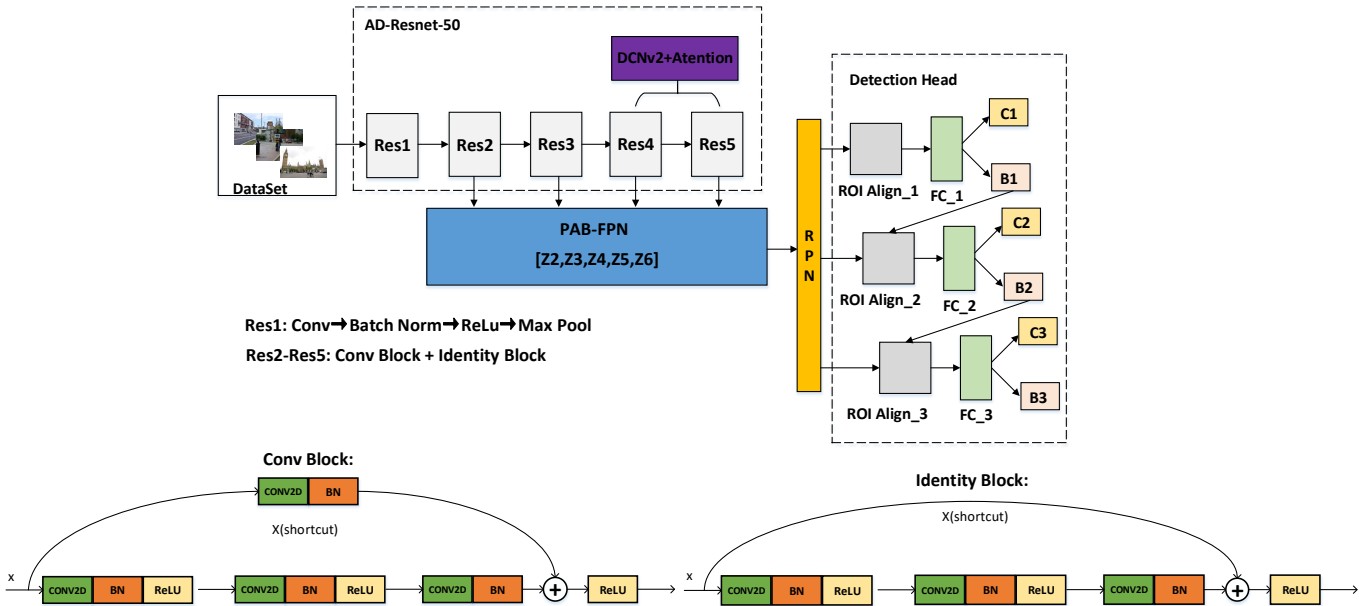

**Figure 1.** The AD-Faster-RCNN model structure.

The loss function of AD-Faster-RCNN is consistent with that of the original algorithm, and it consists of classification loss and regression loss. These two types of loss functions were used in the detection head part of the model. The category loss and location loss were calculated according to the model output and ground truth, and the model was guided to match the ground truth as much as possible in category and location to obtain the best detection results.

The classification function is a two-class cross-entropy loss function, and the regression loss function uses smooth L1 loss [3].

$$smooth_{L_1}(x) = \begin{cases} 0.5x^2 & \text{if} |x| < 1 \\ |x| - 0.5 & \text{otherwise} \end{cases} \tag{1}$$

In the formula, $x$ is the difference between the prediction box and the ground truth.

### 3.2. Backbone Network Design

In this paper, ResNet-50 is used as the basic network; the ideas of deformable convolution and a spatial attention mechanism were used to improve it, and the improved network was named AD-ResNet-50.

#### 3.2.1. Deformable Convolution Suitable for Autonomous Driving Scenarios

In the autonomous driving scene, where the object scale is different and the shape is changeable, the detection of objects using traditional convolutional neural networks is not efficient enough because of the inherent geometric structure (regular sampling point) of the convolutional neural network. The deformable convolution used in this paper modifies the ordinary convolution and improves the deformation modeling ability of the CNN. The basic idea is to learn an offset for the sampling point so that the convolution kernel focuses on the region of interest or object instead of sampling at a fixed location. The comparison between normal convolution sampling and deformable convolution sampling is shown in Figure 2.

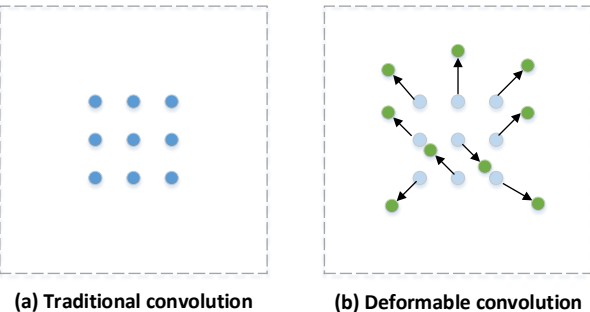

(a) Traditional convolution      (b) Deformable convolution

**Figure 2.** Comparison of ordinary convolution and deformable convolution.

The calculation method of the traditional convolution structure is as follows:

$$y(p_0) = \sum_{p_n \in R} w(p_n) x(p_0 + p_n) \tag{2}$$

In the formula, $w$ represents the weight of each sampled value, $p_n$ is the output position of the convolution, and $p_0$ corresponds to an integer offset to provide nearby semantic information.

Deformable convolution adds a small offset $\Delta p_n$ to traditional convolution, which is calculated by another convolution and can be taken to a decimal offset. The updated convolution calculation method is as follows:

$$y(p_0) = \sum_{p_n \in R} w(p_n) x(p_0 + p_n + \Delta p_n) \tag{3}$$

In the formula, $\Delta p_n$ represents the offset of each sampling point.

After increasing the decimal offset, the corresponding position cannot be found directly in the image of the upper layer during the convolution calculation. The application needs to obtain the corresponding value through methods such as bilinear interpolation.

Since deformable convolution breaks the shape of the conventional sampling area, the sampling points may be expanded to the part outside the area of interest during the use of the model, and more irrelevant information and context information may be included, which would affect the performance of the model. Therefore, deformable convolution v2 [24] proposes an improved scheme. The parallel network not only learns the offset value of each position but also learns the weight of each sampling point to avoid the influence of extreme sampling points on network feature extraction. Through weight control, the influence of having too much context information can be effectively reduced, a greater degree of freedom can be added, and the weights of sampling points that may not be needed can be learned to bring them to zero. The calculation formula then becomes:

$$y(p_0) = \sum_{p_n \in R} w(p_n) x(p_0 + p_n + \Delta p_n) \cdot \Delta m_n \tag{4}$$

In the formula, $\Delta m_n$ represents the weight of each offset point.

Through the parallel convolutional neural network, the sampling point offset value ($\Delta p_n$) and weight value ($\Delta m_n$) can be incorporated into the network learning process. Assuming that the input feature map of the parallel network has N channels, and the sampling point offset part corresponds to the offset value of the two dimensions, then the number of output channels corresponds to 2 N. Meanwhile, the weight network is the weight value of each sampling point, and the number of channels corresponds to the number of input channels, N. The block diagram of its realization is shown in Figure 3.

In the application of this paper, if the deformable convolutional design network is adopted, it will cause problems such as having to deal with a large number of parameters

and difficulty in training the network. To reduce the waste of computing resources, deformable convolution is applied only in the last two stages of the backbone network in the experimental process of this paper. A good balance is thus achieved between parameter quantity and feature extraction.

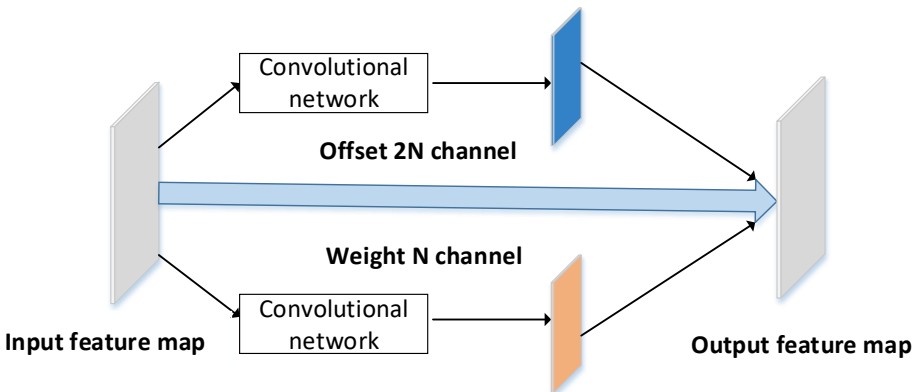

**Figure 3.** Realization of deformable convolution.

### 3.2.2. Spatial Attention Mechanism

The attention mechanism is divided into channel attention and spatial attention. Considering the amount of parameters and the detection speed of the model, this paper used spatial attention, and it was applied only to the last two stages of the backbone network. Spatial attention is used to accurately locate object features in space. In the object detection data set, the proportion of small-object pixels is small [25]. Adding spatial domain attention can accurately locate small objects and improve the accuracy of detection. The calculation formula of the spatial attention module is as follows:

$$M_s(F) = \sigma(f^{7*7}([AvgPool(F), MaxPool(F)])) = \sigma(f^{7*7}([F_{avg}^S; F_{max}^S])) \tag{5}$$

In the formula, $\sigma$ represents the sigmoid function, $F_{avg}^S$ and $F_{max}^S$ represent the output characteristics of global average pooling and global maximum pooling, respectively, and the convolution layer uses a $7 \times 7$ convolution kernel.

The realization of the spatial attention module is shown in Figure 4. First, average pooling and maxpooling are used to compress the input feature map F, and mean and max operations are performed on the input features in the channel dimension. Then, the two obtained feature maps are spliced according to the channel dimensions, and a convolution operation is used to reduce the dimension to one channel. It is ensured that the obtained feature map is consistent with the input feature map in the spatial dimension; finally, the spatial attention feature Ms is generated through the sigmoid function.

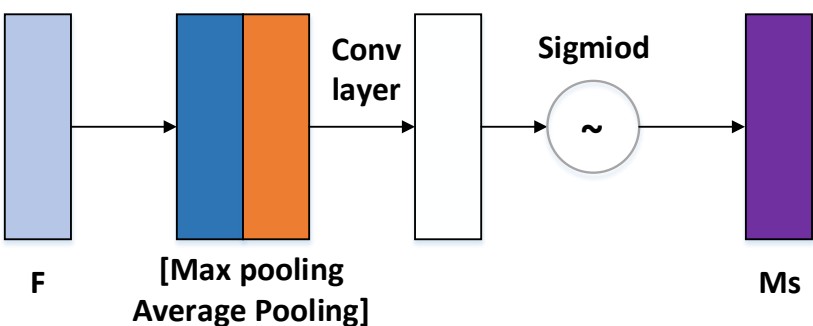

**Figure 4.** Spatial attention module.

The method of adding spatial attention to the structural blocks of the ResNet-50 network is shown in Figure 5. The feature map generated in the previous layer is subjected

to a convolution calculation to generate the input feature map F. After F passes through the spatial attention module, the spatial attention feature Ms is obtained. The elementwise multiplication of F and Ms is performed to obtain a new feature map, $F^1$. Then, $F^1$ and F are added together, the residual module of ResNet is retained, and the generated feature map, $F^2$, is used as the input of the next module.

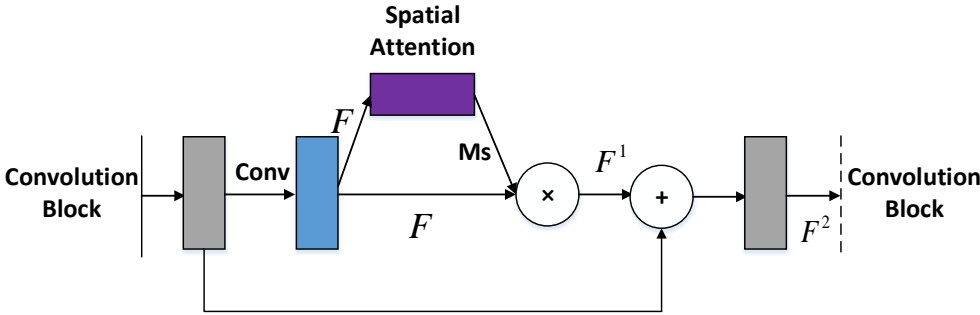

**Figure 5.** ResNet module with attention added.

### 3.3. Feature Fusion Module Design

In the original Faster-RCNN algorithm, the backbone network extracts only the high-level features of the last layer and sends them to the RPN to generate the region of interest, ignoring position and detail information at the bottom layer, which results in the poor detection of small objects. Therefore, later generations proposed a top-to-bottom feature pyramid structure (FPN) combined with Faster-RCNN to transfer high-level semantic information and fuse the features of different scales.

However, there are still shortcomings: (1) only semantic information is enhanced, and positioning information is not transmitted; (2) these top-to-bottom and bottom-to-top methods pay more attention to adjacent resolutions and nonadjacent layers. The semantic information contained will be diluted in the process of information fusion.

In response to these issues, this paper proposes a new feature fusion method of path aggregation-balanced FPN (PAB-FPN) based on an FPN, as shown in Figure 6. First, the feature maps C2–C5 generated by ResNet-50 are merged through a top-down feature pyramid to obtain feature maps P2–P5, and P6 is obtained by two downsamplings from P5 to enhance robustness. Then, bottom-up pyramid fusion is added behind the FPN, and the strong positioning feature at the bottom layer is transferred to obtain the feature map N2–N6. Then, the five-level features are uniformly scaled to the N4 size, and the integration operation is performed. The specific operation formula is as follows:

$$C = \frac{1}{L} \sum_{l=l_{\min}}^{l_{\max}} C_l \tag{6}$$

After passing the non-local [26] module refinement, the obtained feature map is scaled to the original size and added to the original feature in order to enhance the original feature and obtain Z2–Z6. In this process, feature maps of each scale can obtain equal information from other feature maps, making the process of feature fusion more balanced.

### 3.4. Detection Head Design

#### 3.4.1. Cascade Detector

In Faster-RCNN, the IOU threshold is usually used to define positive and negative samples. The threshold of the training detector defines the quality of detection. A threshold that is too low will result in low detection quality, but as the threshold increases, the detection performance tends to decrease. There are two reasons for this: (1) The reduction in the number of positive samples can easily lead to overfitting. (2) When the proposal threshold output by the RPN is too far from the threshold set by the trainer, there is a mismatch problem. In response to this, and borrowing the idea of Cascade-RCNN,

we used three cascade detectors, as shown in the "head" part of Figure 1, where each detector included ROI Align, a fully connected layer FC, a classification score C, and border regression B.

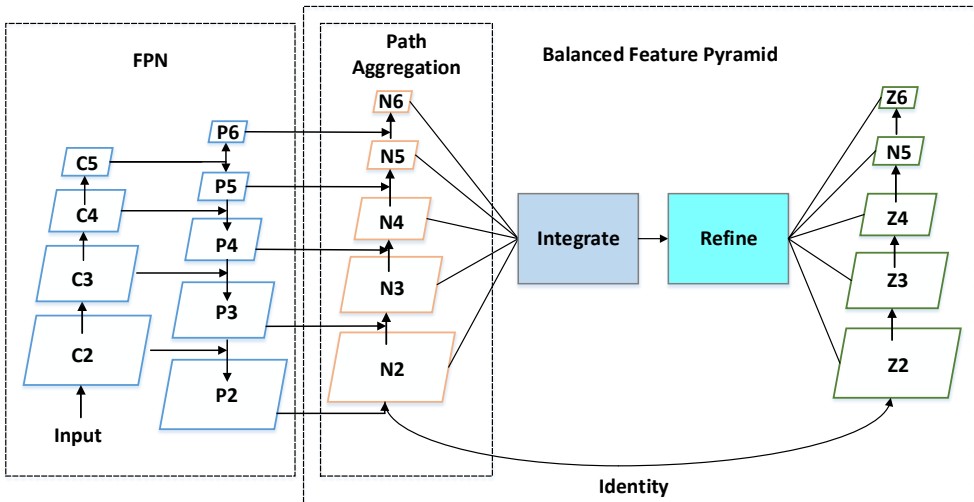

**Figure 6.** PAB-FPN network structure.

In the training stage, the RPN network puts forward about 2000 proposals, which are sent to the cascaded detection head structure. First, the IOU between each proposal and the ground truth is calculated and divided into positive and negative samples. The candidate target area is then resampled through the border regression B output by the previous detector, and the IOU threshold (0.5, 0.6, 0.7) is gradually increased in order to train it to obtain a new classification score, C, and border regression, B; consequently, each detector can focus on the proposal of IOU in a certain range, ultimately improving the sample quality and network training effect.

### 3.4.2. Side-Aware Boundary Localization

The mainstream frame generation method is shown in Figure 7a, which obtains the frame information by predicting the center point and the center offset; however, due to the large variance of the regression object, this does not improve the positioning accuracy very much. This paper draws on the idea of side-aware boundary localization (SABL) [27], as shown in Figure 7b, and locates the border of the bbox by using the border information of the feature map content to obtain higher-quality bbox border information.

The specific steps are as follows: (1) divide the object space into multiple buckets, (2) determine which bucket the border of the object is in, and (3) return to the offset of the border line from the center of the bucket. In this way, high-quality frame coordinate information can be obtained through the accurate regression of the four frames. This frame positioning method is applied to three cascade detectors to improve the accuracy of detection.

### 3.5. Soft Non-Maximum Suppression

NMS [28] is an algorithm for removing non-maximum values, which can remove the repeated detection frame in the object detection task and find the best object detection position. In the model training process, the NMS algorithm is used to post-process a large number of generated candidate frames, and redundant candidate frames are removed to obtain the most representative results to speed up and increase the efficiency of object detection and improve detection accuracy. The feature of Soft-NMS [29] is that it can be re-evaluated recursively according to the current score instead of crudely zeroing, which allows it to avoid missed detection when similar objects have high overlap. At the same

time, using this algorithm does not necessitate the retraining of the model and will not increase the training overhead.

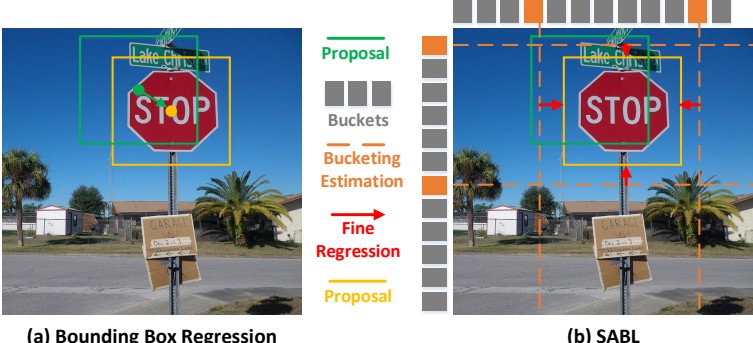

**Figure 7.** Side-Aware Boundary Localization.

## 4. Experiments and Results

### 4.1. Data Set and Evaluation Indicators

This paper used MS COCO2017 [30] with 80 categories as the experimental data set. To prove the effectiveness of the AD-Faster-RCNN model proposed in this paper in autonomous driving scenarios, we selected the most common pedestrians, cars, buses, trucks, and bicycles. The eight categories of motorcycles, traffic lights, and stop signs were used as experimental data. There were a total of 118,287 training sets and 5000 test sets. Compared with other data sets, the pictures in COCO included natural pictures and common-object pictures from real life. The backgrounds were more complicated, the numbers of objects were larger, and the object sizes were smaller. Therefore, the tasks used on the COCO data set were more difficult. For the detection task, the standard for measuring the quality of a model is more likely to use the detection results on the COCO data set.

To verify the reliability and stability of the model, after the training was completed, 5000 images in the test set were recognized. We chose the average precision (AP) as the evaluation index of the validity of the test results. The AP in the COCO data set did not use a fixed IOU threshold but averaged multiple IOU thresholds. The threshold was between 0.5 (coarse positioning) and 0.95 (perfect positioning). We used the number of frames per second (FPS) to evaluate the detection speed of the network.

### 4.2. Experimental Environment and Experimental Plan

This paper used PyTorch and the mmdetection open-source framework provided by the Chinese University of Hong Kong to conduct experiments under the Ubuntu 16.04 system. The machine configuration used in the experiment was: an Intel(R) Core(TM) i7-7800X CPU, 32 G memory, and 2 NVIDIA 1080Ti GPUs (11 GB). The software environment used was PyTorch 1.5.0, Python 3.6, CUDA 10.1, and cuDNN 7.3.1.

When training the model, we first loaded the pretraining weights of ResNet-50 on ImageNet to speed up the model convergence, and we used stochastic gradient descent (SGD) to optimize the loss function. The initial learning rate was set to 0.005, and the warm-up learning rate [18] method was used. A smaller learning rate was used at the beginning of training. When the model stabilized, we selected the preset learning rate for training. The momentum was set to 0.9, the weight decay coefficient was set to 0.0001, the batch size was set to 2; a total of 12 epochs were trained, and the learning rate was set to 0.0005 and 0.00005 for the 9th and the 12th epochs, respectively, with one epoch for each iteration. We saved the model once and finally selected the model with the highest accuracy. Using the multiscale training enhancement method, the input picture size was set to 1333 × 640 and 1333 × 800, and each picture was randomly selected for training at a scale that improved the robustness of the model.

*4.3. Analysis of the Experimental Results*

4.3.1. Comparison of Faster-RCNN with the Results of This Algorithm

To verify the effectiveness of the improved algorithm in this paper, the baseline used Faster-RCNN with ResNet-50 as the backbone network, and then added an FPN structure. Table 1 shows the statistics of the object detection results of the Faster-RCNN and AD-Faster-RCNN algorithms on the data set.

**Table 1.** AD-Faster-RCNN object detection results.

| Method | AP | Person | Car | Bus | Truck | Bicycle | Motorcycle | Traffic Light | Stop Sign |
|---|---|---|---|---|---|---|---|---|---|
| Faster-RCNN | 0.431 | 0.538 | 0.423 | 0.593 | 0.315 | 0.270 | 0.385 | 0.270 | 0.650 |
| AD-Faster-RCNN | 0.508 | 0.620 | 0.503 | 0.706 | 0.408 | 0.350 | 0.475 | 0.297 | 0.702 |

The AD-Faster-RCNN in this paper achieved 50.8% AP (IOU = 0.5:0.95) among the eight types of objects in the COCO2017 data set—pedestrians, cars, buses, trucks, bicycles, motorcycles, traffic lights, and stop signs—which is an increase of 7.7% compared to the baseline; in particular, for the two categories of buses and bicycles, the accuracy increased by 11.3% and 12%, respectively. For bicycles and traffic lights, the performance of the two models was not very good because the amount of data was too low and the objects were too small, which is not conducive to feature extraction and expression.

To see the detection effect more intuitively, Figure 8 shows the visualization of the results of detecting vehicles and pedestrians in common scenes such as roads and streets with the use of the Faster-RCNN and AD-Faster-RCNN algorithms. Figure 8(a1)–(a4) show the detection result of the Faster-RCNN algorithm. In the figure, it can be seen that there is a certain degree of leak detection for small objects and occluded objects (marked with a red circle). Figure 8(b1)–(b4) show the AD-Faster-RCNN algorithm detection results. Compared with the Faster-RCNN algorithm, AD-Faster-RCNN has better detection performance on small-scale pedestrians and vehicles, the score is generally higher, and it has higher detection efficiency.

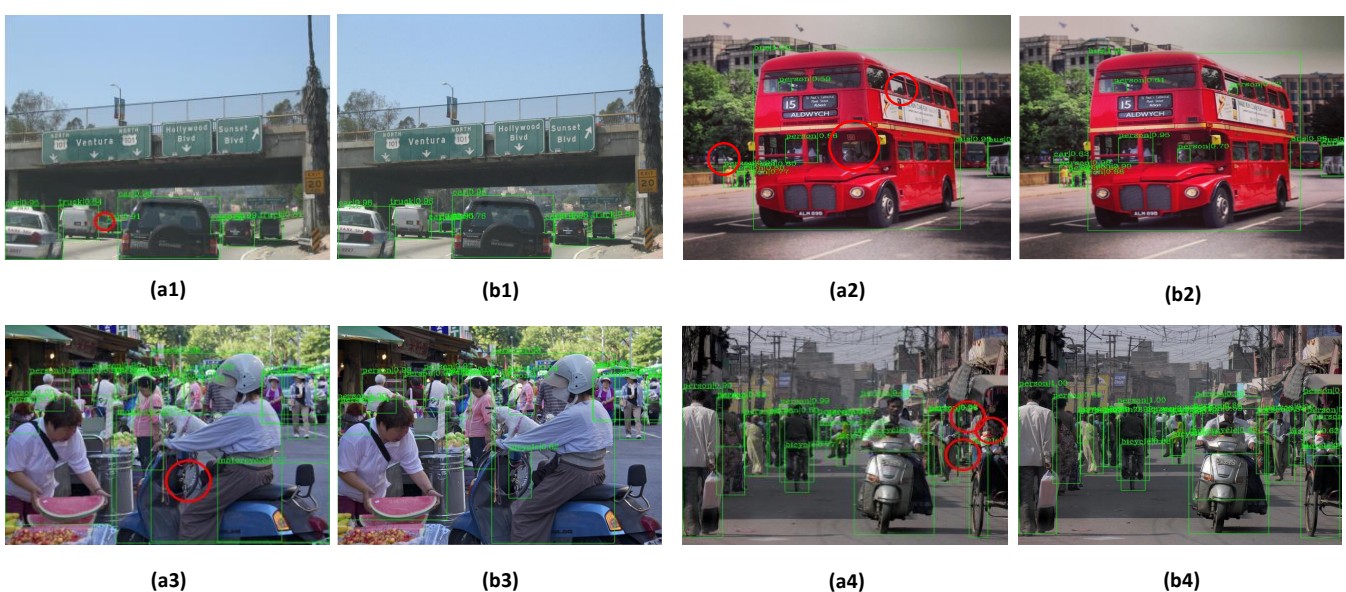

| | | | |
|---|---|---|---|
| **(a1)** | **(b1)** | **(a2)** | **(b2)** |
| **(a3)** | **(b3)** | **(a4)** | **(b4)** |

**Figure 8.** Comparison of the visualization results.

4.3.2. Ablation Experiment

Ablation experiments are a commonly used experimental method in the field of deep learning and are mainly used to analyze the influence of different network branches on the entire model. To further analyze the influence of the deformable convolution, spatial attention mechanism, optimized multiscale feature fusion, cascade detection head,

side border positioning, and soft non-maximum suppression on the Faster-RCNN model proposed in this paper, an ablation experiment was performed. The specific results are shown in Table 2, where "✓" represents the corresponding improvement method.

It can be seen from the table that regardless of the IOU threshold, each method improves the accuracy of the model to a certain extent; in particular, when the IOU threshold is higher, it has better performance. Among the methods, deformable convolution, cascade detection heads, and side-aware boundary localization greatly improve the accuracy of the model, increasing it by 2.7%, 2.8%, and 2.1%, respectively, but due to the increase in the number of calculations, the FPS is reduced by 1–2.

The spatial attention mechanism, optimized multiscale feature fusion, and soft non-maximum suppression, at the expense of a small number of FPS, improved the accuracy of the model by 0.7%, 0.8%, and 0.8%, respectively. The experiments proved that the improved strategy proposed in this paper for Faster-RCNN is meaningful for improving the object detection efficiency in autonomous driving scenarios.

**Table 2.** Results of the ablation experiments.

| Method | DCNv2 | Attention | PAB-FPN | Cascade Head | SABL | Soft-NMS | AP | AP50 | AP75 | FPS |
|---|---|---|---|---|---|---|---|---|---|---|
| Faster-RCNN | | | | | | | 0.431 | 0.674 | 0.447 | 12.5 |
| Improve1 | ✓ | | | | | | 0.458 | 0.706 | 0.484 | 11 |
| Improve2 | | ✓ | | | | | 0.438 | 0.684 | 0.451 | 12 |
| Improve3 | | | ✓ | | | | 0.439 | 0.686 | 0.466 | 11.2 |
| Improve4 | | | | ✓ | | | 0.459 | 0.680 | 0.491 | 11 |
| Improve5 | | | | | ✓ | | 0.452 | 0.672 | 0.477 | 10.3 |
| Improve6 | | | | | | ✓ | 0.439 | 0.676 | 0.462 | 12.4 |
| AD-Faster-RCNN | ✓ | ✓ | ✓ | ✓ | ✓ | ✓ | 0.508 | 0.712 | 0.541 | 6 |

### 4.3.3. Comparison of Different Algorithms

To further prove the effectiveness and scientificity of the algorithm in this paper, it is compared with the current mainstream two-stage object detection algorithm. The comparison results are shown in Table 3.

**Table 3.** Comparison of the algorithm in this paper with other algorithms.

| Method | Backbone | AP | AP50 | AP75 | APs | APm | APl | FPS | GFlOPs |
|---|---|---|---|---|---|---|---|---|---|
| Faster-RCNN | ResNet-50-FPN | 0.431 | 0.672 | 0.447 | 0.262 | 0.463 | 0.623 | 12.5 | 230.1 |
| Libra-RCNN | ResNet-101 | 0.444 | 0.680 | 0.461 | 0.278 | 0.475 | 0.639 | 12.3 | 238.5 |
| Cascade-RCNN | ResNet-101 | 0.462 | 0.681 | 0.486 | 0.282 | 0.493 | 0.660 | 11.1 | 258.3 |
| TridentNet | ResNet-101 | 0.467 | 0.694 | 0.488 | 0.284 | 0.504 | 0.674 | 5.7 | 271.3 |
| **AD-Faster-RCNN** | **AD-ResNet-50** | **0.508** | **0.712** | **0.541** | **0.313** | **0.549** | **0.723** | **6** | **287.6** |

It can be seen from Table 3 that the AP value of the AD-Faster-RCNN algorithm proposed in this paper in complex scenes is increased by 7.7%, 6.4%, 4.6%, and 4.1% as compared with Faster-RCNN, Libra-RCNN, Cascade-RCNN, and TridentNet [31], respectively; the FPS is reduced by 6.5, 6.3, and 5.1, which is basically the same as that of TridentNet. Due to the larger model, GFLOPs are also higher than the algorithms above by 57.5, 49.1, 29.3, and 16.3, respectively. When the IOU threshold is equal to 50 and 75, the detection accuracy is the highest, especially when the IOU threshold is equal to 75; the accuracy is also much higher than that of the other models, which reflects the superiority of the algorithm in this paper under strict conditions. At the same time, AD-Faster-RCNN has good detection results for small objects (an area less than 32), medium objects (an area greater than 32 and less than 96), and large objects (an area greater than 96). Compared with other algorithms, its detection accuracy is the highest. This also proves that the method proposed in this paper can address the problem of misdetection and missed detection of small and occluded

objects. In general, the algorithm in this paper is meaningful for improving the effect of object detection in autonomous driving scenarios.

### 4.3.4. Experiments on Autonomous Driving Scenarios

In order to verify the effectiveness of the AD-Faster-RCNN proposed in the paper in the autonomous driving scenario, the model was experimentally verified on the BDD100k data set. The validation set of BDD100k contained 10,000 pictures. First, we converted it to COCO format, and then verified the trained model on the validation set. The experiments showed that the detection accuracy of AD-Faster-RCNN proposed in this paper is higher than that of the baseline in each category in the autonomous driving scene.

As can be seen from Table 4, on the BDD100k data set, AD-Faster-RCNN on AP compared to Faster-RCNN, Libra-RCNN, Cascade-RCNN, and TridentNet increased by 4.1%, 3.2%, 2.6%, and 2.3%, respectively. Especially in the two categories of Person and Truck, there was a big improvement.

Compared with the COCO data set, the overall accuracy of the BDD100k data set was lower, which may be because the scene was more complex and that there were mostly small objects, causing the detection efficiency to be not as good as that of the COCO data set.

Figure 9(a1)–(a4),(b1)–(b4) are the detection results of Faster-RCNN and AD-Faster-RCNN on the BDD100k data set, respectively. It can be seen from the figure that AD-Faster-RCNN solves the problem of missed detection of small objects to a certain extent. This includes detection in various driving scenarios such as night, rain, street, and road. This shows that the model proposed in this paper has a certain generalization ability and versatility, and can effectively adapt to object detection in autonomous driving scenarios.

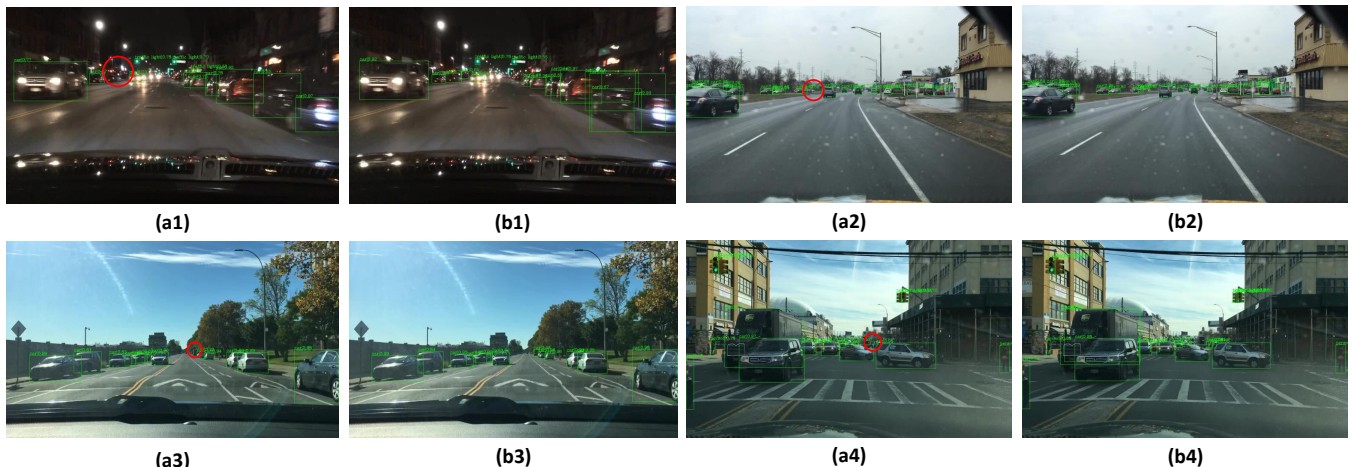

**(a1)** **(b1)** **(a2)** **(b2)**

**(a3)** **(b3)** **(a4)** **(b4)**

**Figure 9.** Visualization results of BDD100K.

**Table 4.** BDD100k experimental comparison.

| Method | AP | Bike | Bus | Car | Motor | Person | Traffic light | Stop sign | Truck |
|---|---|---|---|---|---|---|---|---|---|
| Faster-RCNN | 0.372 | 0.253 | 0.321 | 0.632 | 0.169 | 0.447 | 0.365 | 0.524 | 0.261 |
| Libra-RCNN | 0.381 | 0.262 | 0.325 | 0.641 | 0.171 | 0.468 | 0.374 | 0.529 | 0.281 |
| Cascade-RCNN | 0.387 | 0.268 | 0.328 | 0.650 | 0.172 | 0.472 | 0.381 | 0.531 | 0.295 |
| TridentNet | 0.390 | 0.274 | 0.329 | 0.652 | 0.173 | 0.477 | 0.385 | 0.531 | 0.302 |
| AD-Faster-RCNN | 0.413 | 0.296 | 0.334 | 0.668 | 0.176 | 0.512 | 0.403 | 0.537 | 0.379 |

### 4.3.5. Natural Scene Video Detection Experiment

To verify the effect of the algorithm in object detection in natural scenes, a test was carried out with video. The experimental results are shown in Figure 10. The video was downloaded from the Internet, and the video size was 1024 × 1024. The algorithm in this paper can effectively detect the objects in the video as a whole. Compared with Faster-

RCNN, the AD-Faster-RCNN proposed in this paper achieved a certain improvement in the detection of small objects in natural scenes, but there is still room for improvement in the detection speed in future work.

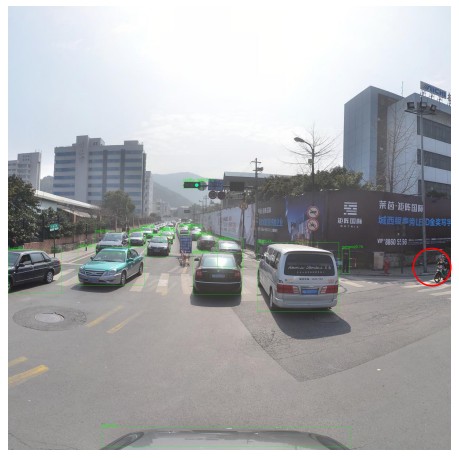 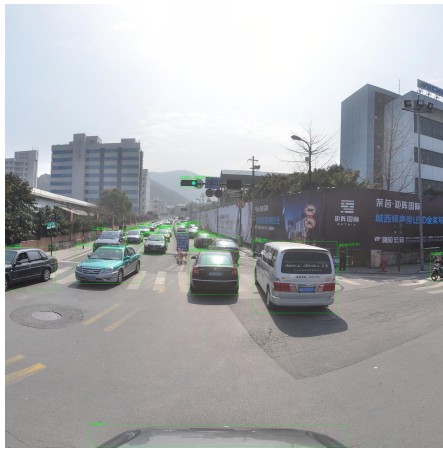

(a) Faster-RCNN · · · · · · · · · · · · · · · · · · · · · · · (b) AD-Faster-RCNN

**Figure 10.** Video detection effect.

## 5. Conclusions

Based on the Faster-RCNN algorithm, this paper proposed the AD-Faster-RCNN algorithm to overcome the problems of occlusion and dense and small objects in object detection tasks in autonomous driving scenarios by improving the backbone network, the feature fusion module, and the detection head. The experimental results showed that the algorithm proposed in this paper could effectively improve the accuracy of object detection tasks in autonomous driving scenarios. Lightening the model and balancing the detection accuracy and speed is a direction that needs to be studied as the next step.

**Author Contributions:** Conceptualization, Y.Z., S.W. and D.W.; methodology, Y.Z. and S.W.; validation, Y.Z., S.W., D.W., J.M. and I.R.; investigation, S.W. and D.W.; visualization, Y.Z. and I.R.; writing—original draft preparation, Y.Z., S.W., D.W., J.M. and I.R.; writing—review and editing, Y.Z., S.W., D.W. and J.M. All authors have read and agreed to the published version of the manuscript.

**Funding:** National Natural Science Foundation of China: 61773330.

**Institutional Review Board Statement:** Not applicable.

**Informed Consent Statement:** Not applicable.

**Data Availability Statement:** The data presented in this study are available on request from the corresponding author.

**Acknowledgments:** This work was supported by the National Natural Science Foundation of China (61773330), a project of the Hunan National Applied Mathematics Center (2020YFA0712503), a project of the Shanghai Municipal Science and Technology Commission (19511120900), and the Research Project of the Department of Education of Hunan Province (19C1740), the Hunan Province Science and Technology Plan project (2020GK2036), and the Aeronautical Science Foundation of China (20200020114004).

**Conflicts of Interest:** The authors declare no conflict of interest.

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
