# Peer review of "Object Detection in Autonomous Driving Scenarios Based on an Improved Faster-RCNN"

_applsci, doi:10.3390/app112411630_

Round 1

Reviewer 1 Report

Overview:

The article proposes a Faster-RCNN architecture that is specialized to autonomous driving scenarios – AD-Faster-RCNN.

The article presents the architecture of the system in Figure 1 but immediately references Figure 5 and in the meantime the architectural feature – the reliance to the ResNet-50 – is not mentioned. The coming sentences – detailing the Res1 to Res5 – are again unclear.

Several architectural details from Figure 1 are unclear, e.g., the “Cascade Head” and the interaction with the training of the parameter inference part of the system.

Simultaneously, the article presents a lot of features, like the “smooth L1 loss” – on line 137 – or the deformable convolution – line 142 – that _could_ help in improving the performance of a CNN system; however, the details are missing, and one could ask the following questions: (1) what exactly is the smooth L1 loss, (2) how the deformable convolution is learned.

Since these incipient notions are unclear, the rest of the article is unclear also. For example, in equation 3 there is a “\Delta m_n” that is never explained and one can only guess that is related to the “greater degree of freedom” mentioned in line 155.

The lack of mathematical definitions is a heavily missing block from the article.

The results in Table 3 are clearly encouraging – showing that the newly built method has improved characteristics related to performance but it also highlights the computational costs at which these advances are achieved.

To sum up: given the unclarities in defining the method, I cannot recommend its publication.

Author Response

Point 1:

The article presents the architecture of the system in Figure 1 but immediately references Figure 5 and in the meantime the architectural feature – the reliance to the ResNet-50 – is not mentioned. The coming sentences – detailing the Res1 to Res5 – are again unclear.

Response 1:

When introducing the system architecture of Figure 1, the reference to Figure 5 was deleted; the text introduction to the backbone network Resnet-50 was added, and the description of Resnet-50 in Figure 1 (the structure of Res1-Res5) was added; introduction An introduction to the role of Resnet-50 in the overall system framework (the process of extracting four different levels of features and passing them into PAB-FPN).

Point 2:

Several architectural details from Figure 1 are unclear, e.g., the “Cascade Head” and the interaction with the training of the parameter inference part of the system.

Response 2:

Each module in Figure 1 is described in detail in the corresponding section, and the principle of the cascade detection head is further introduced in Section 3.4.1. Its principle is to connect three cascaded detection heads in series, and set the thresholds of the three detection heads to 0.5, 0.6, and 0.7 respectively. The output of the previous detection head is used as the input of the next detection head, so that each detection head can focus on the detection within a certain threshold, thereby obtaining a higher quality detection frame.

Point 3:

Simultaneously, the article presents a lot of features, like the “smooth L1 loss” – on line 137 – or the deformable convolution – line 142 – that _could_ help in improving the performance of a CNN system; however, the details are missing, and one could ask the following questions: (1) what exactly is the smooth L1 loss, (2) how the deformable convolution is learned.

Response 3:

The smooth L1 loss is proposed by the previous generation of Fast-RCNN, and references have been added, and the loss function of the Baseline in this article is also a smooth L1 loss without any improvement, so it is just a brief introduction. In the paper, the formula and introduction of smooth L1 loss are newly added; the method for implementing deformable convolution (text introduction and picture) is added --- learning the offset and the weight of each sampling point through two parallel convolutional networks .

Point 4:

Since these incipient notions are unclear, the rest of the article is unclear also. For example, in equation 3 there is a “\Delta m_n” that is never explained and one can only guess that is related to the “greater degree of freedom” mentioned in line 155.The lack of mathematical definitions is a heavily missing block from the article.

Response 4:

The description of "\Delta m_n" has been added, which is the weight mentioned in Response 3.

Point 5:

The results in Table 3 are clearly encouraging – showing that the newly built method has improved characteristics related to performance but it also highlights the computational costs at which these advances are achieved.

Response 5:

The focus of this paper is on the detection of small objects in autonomous driving scenarios, so the calculation cost will increase to a certain extent, which will be the next research direction. Thank you very much for your advice! I will continue to work hard.

Reviewer 2 Report

One of the most important issues when moving autonomous vehicles is ensuring their safety. The algorithms used to recognize small and obscured objects can be unreliable. For this reason, modification of the algorithms used is very important.

In the reviewed paper, the authors proposed an improved Faster-RCNN algorithm. They carried out an experimental study and, based on the results obtained, showed that it is more accurate than other algorithms used.

I therefore believe that the article should be published.

The literature selected by the authors is current and appropriately chosen.

Author Response

Response

Thank you very much for your praise of my paper!I will continue to work hard to modify it to make it better.

Reviewer 3 Report

The paper focuses on the object detection systems that are an essential part of autonomous driving systems. A new approach is proposed that improved the performances of object detection (compared with similar solutions), yet, as the authors also state, further work is needed.

Some technical improvements are needed. The references list needs to be updated according to the journal requirements. Currently, different styles are applied. The authors should mention/comment on all references at least once in the paper. Now, several references are just listed but never mentioned in the paper. In the text, the references should be numbered consecutively.

Similar to the references list, all figures should be commented on at least once (Fig 7, those (a) and (b) cases). However, it is unusual to see multiple 'a' and 'b' sub-figures on the same figure. The location of tables and figures is sometimes confusing, i.e., they do not always follow the text.

The authors used MS COCO dataset for the model development and validation. Since we are talking about the autonomous driving context, do you think another data set, more focused on traffic images, would make more sense?

I would also like to see a discussion about the algorithm's speed and computer power needed, compared to others. The concluding chapter should be improved since the abstract and the conclusion are two chapters that can 'sell' the paper to the potential reader.

Author Response

Point 1:

Some technical improvements are needed. The references list needs to be updated according to the journal requirements. Currently, different styles are applied. The authors should mention/comment on all references at least once in the paper. Now, several references are just listed but never mentioned in the paper. In the text, the references should be numbered consecutively.

Response 1:

The reference list has been updated according to journal requirements; all references have been listed in the paper, and have been mentioned or evaluated; references in the main text have been numbered consecutively. 

Point 2:

Similar to the references list, all figures should be commented on at least once (Fig 7, those (a) and (b) cases). However, it is unusual to see multiple 'a' and 'b' sub-figures on the same figure. The location of tables and figures is sometimes confusing, i.e., they do not always follow the text.

Response 2:

All figures in the paper have been commented, and (a) , (b) in Figure 8 (original figure 7) have been modified to (a1)-(a4), (b1)-(b4); the positions of tables and pictures have been modified.

Point 3:

The authors used MS COCO dataset for the model development and validation. Since we are talking about the autonomous driving context, do you think another data set, more focused on traffic images, would make more sense?

Response 3:

It is indeed more meaningful to choose a data set that focuses on traffic. We have newly added the experimental results (Table 4) of the autonomous driving scene data set BDD100k data set, as well as the visualization results and comparison (Figure 9). And achieved good results, indicating that the model we proposed has relatively good generality.

Point 4:

I would also like to see a discussion about the algorithm's speed and computer power needed, compared to others. The concluding chapter should be improved since the abstract and the conclusion are two chapters that can 'sell' the paper to the potential reader.

Response 4:

FPS is the speed of the algorithm, and the comparison of GFLOPs (Giga Floating-point Operations Per Second) has been added in Table 3, and made a comparison; Some changes have been made to the conclusions. Thank you very much for your advice on my paper! I will continue to work hard.

Reviewer 4 Report

The authors present a series of improvements on a Faster-RCNN detection-regression type network. Namely, they introduce the deformable convolutions of [26] in the backbone (a ResNet 50 architecture), in the last two layers.  An attention mechanism is also included in the residual network blocks. Additionally, they include SABL (side-aware boundary localization) from Wang et al. 2020 [ArXiv], and a cascaded network "head", along soft non-maximum suppression (NMS). This leads to a 7.7% improvement in AP from the baseline (43.1 vs. 50.8) for the considered classes in the COCO dataset.

However, I have certain concerns regarding the novelties that are claimed: in lines 62-70 it is claimed that "deformable convolution and a spatial attention mechanism are INTRODUCED", as if these were a novelty, however these are used in the literature, DCNv2 is referenced as [26] later in the text (not at this point in the text, why?), however "spatial attention" is also found in the literature, and does not seem to be referenced. The same happens with SABL, which is not cited as Wang et al. 2020 [ArXiv], and can be mistakenly taken as original work by the authors. A similar problem is found in lines 67-69: how is FPN new in this context? (e.g. Detectron2 [by Facebook, Inc.] project models, which generally use ResNet50 as a backbone, and are available on GitHub, use FPN in some/all variants).

Furthermore, it is claimed that one-stage algorithms perform worse, even later versions of this family of algorithms. This is however, not justified in the text, by references, result tables, links to benchmarks, or similar (lines 51-56; 73; 94; etc.). I am not claiming the opposite, only the lack of evidence to support this claim.

Finally, regarding the choice for the dataset, it would make much more sense, given the context of application, to use a specialised dataset for autonomous driving (such as KITTI, CityScapes, Apollo, or similar), rather than a hand-picked subset of classes of a general detection-regression challenge dataset, as is COCO. If choosing COCO to prove the point of the superiority of the network, why choose only a few classes? If the selection is justified by the application, why not extend the experiments to KITTI and similar autonomous driving datasets, that are much more likely to resemble the algorithm's day-to-day recognition scenarios? This is also not justified in the text. Pictures from COCO might show STOP signs from the perspective of a human taking a picture while standing, and not from the dashboard of a car (as is the case with these other specialised datasets mentioned above).

To conclude, it is the opinion of this reviewer, that the paper has potential,  and the ideas are good, but the unsubstantiated claims, and dataset choice lead me to conclude this paper is not fit for publication in its current form, needs additional work, a proper rewrite (in special in to justify superiority of the selected network model, the choice of dataset, etc.), as well as properly referencing previous work, citing authors as required, and clearly delimiting what are the authors' contributions, and which parts are taken from others' works. Finally, regarding the dataset, it is recommended to run further experiments with datasets more fit to the task at hand, and if ALSO using general datasets, showing results for all available classes, not just a selection.

Other remarks: is TridentNet referenced in the text? Appears in the reference section, is mentioned in the text, but a cite to [31] is missing? In line 47, most recent reference is from 2018? "redundant frames" is mentioned several times in the text, do you mean "redundant bounding boxes" or "redundant detection instances"? "frames" gives the impression of meaning "video frames" or "images". Line 263 mentions "preheating LR method", reference missing? In line 267 small numbers appear, you could use scientific notation instead (e.g. 5x10^-5 or 5e-5). In line 293 the "better applicability to autonomous driving scenarios" is not justified, since there are no results provided with "real-world" car dashboard datasets (specialised on autonomous driving research). Figure 8 shows images from a video downloaded from the Internet, which seems to be taken from a bridge above the road, rather than the dashboard of a car. It would have been a good opportunity to show such data. In line 342 the statement "algorithm to solve" is too bold ("to overcome some of the issues", or similar would be more adjusted to reality).

Author Response

Point 1:

However, I have certain concerns regarding the novelties that are claimed: in lines 62-70 it is claimed that "deformable convolution and a spatial attention mechanism are INTRODUCED", as if these were a novelty, however these are used in the literature, DCNv2 is referenced as [26] later in the text (not at this point in the text, why?), however "spatial attention" is also found in the literature, and does not seem to be referenced. The same happens with SABL, which is not cited as Wang et al. 2020 [ArXiv], and can be mistakenly taken as original work by the authors. A similar problem is found in lines 67-69: how is FPN new in this context? (e.g. Detectron2 [by Facebook, Inc.] project models, which generally use ResNet50 as a backbone, and are available on GitHub, use FPN in some/all variants).

Response 1:

References to deformable convolution, spatial attention mechanism, SABL, FPN have been added to the paper; the BAP-FPN proposed in this article is an improvement of FPN and one of the innovations of this article (adding a top-direction after FPN The path below transfers the high-level semantic information-rich feature maps, and then resizes the feature maps of all layers to the size of N4, performs an addition and average operation, and finally refines and then resizes to the original size). It is described in detail in section 3.3.

Point 2:

Furthermore, it is claimed that one-stage algorithms perform worse, even later versions of this family of algorithms. This is however, not justified in the text, by references, result tables, links to benchmarks, or similar (lines 51-56; 73; 94; etc.). I am not claiming the opposite, only the lack of evidence to support this claim.

Response 2:

In References 5-13, we can learn that the one-stage object detection algorithm is inferior to the two-stage algorithm in the detection of small objects. In the detection of small objects, yolov4 in 2020 is not as good as Faster-RCNN in 2016. A new explanation is added in line 53 of the paper.

Point 3:

Finally, regarding the choice for the dataset, it would make much more sense, given the context of application, to use a specialised dataset for autonomous driving (such as KITTI, CityScapes, Apollo, or similar), rather than a hand-picked subset of classes of a general detection-regression challenge dataset, as is COCO. If choosing COCO to prove the point of the superiority of the network, why choose only a few classes? If the selection is justified by the application, why not extend the experiments to KITTI and similar autonomous driving datasets, that are much more likely to resemble the algorithm's day-to-day recognition scenarios? This is also not justified in the text. Pictures from COCO might show STOP signs from the perspective of a human taking a picture while standing, and not from the dashboard of a car (as is the case with these other specialised datasets mentioned above).

Response 3:

We have newly added the experimental results (Table 4) of the autonomous driving scene data set BDD100k data set, as well as the visualization results and comparison (Figure 9). And achieved good results, indicating that the model we proposed has relatively good generality.

Point 4:

To conclude, it is the opinion of this reviewer, that the paper has potential,  and the ideas are good, but the unsubstantiated claims, and dataset choice lead me to conclude this paper is not fit for publication in its current form, needs additional work, a proper rewrite (in special in to justify superiority of the selected network model, the choice of dataset, etc.), as well as properly referencing previous work, citing authors as required, and clearly delimiting what are the authors' contributions, and which parts are taken from others' works. Finally, regarding the dataset, it is recommended to run further experiments with datasets more fit to the task at hand, and if ALSO using general datasets, showing results for all available classes, not just a selection.

Response 4:

Previous work has been cited and all references in the list have been cited; the author’s contribution is to propose the AD-Faster-RCNN algorithm for the detection of small objects in autonomous driving scenarios; improve FPN and propose PAB-FPN; Borrowing from the deformable convolution and spatial attention mechanism, proposed the AD-Resnet-50. Finally, the experimental verification on the BDD100K data set is added.

Point 5:

Other remarks: is TridentNet referenced in the text? Appears in the reference section, is mentioned in the text, but a cite to [31] is missing? In line 47, most recent reference is from 2018? "redundant frames" is mentioned several times in the text, do you mean "redundant bounding boxes" or "redundant detection instances"? "frames" gives the impression of meaning "video frames" or "images". Line 263 mentions "preheating LR method", reference missing? In line 267 small numbers appear, you could use scientific notation instead (e.g. 5x10^-5 or 5e-5). In line 293 the "better applicability to autonomous driving scenarios" is not justified, since there are no results provided with "real-world" car dashboard datasets (specialised on autonomous driving research). Figure 8 shows images from a video downloaded from the Internet, which seems to be taken from a bridge above the road, rather than the dashboard of a car. It would have been a good opportunity to show such data. In line 342 the statement "algorithm to solve" is too bold ("to overcome some of the issues", or similar would be more adjusted to reality).

Response 5:

TridentNet has been cited; the latest reference is 2020; "redundant frame" refers to "redundant bounding boxes", which has been modified; the learning rate warm-up method first appeared in the Resnet literature [18], and the citation has been added; the decimal has been changed Into scientific notation; a new data set was added, and the detection video was replaced (from the perspective of the car dashboard) ; the statement of "solving algorithm" was revised to "can overcome the problem of poor detection of small objects".

Thank you very much for your advice!

Round 2

Reviewer 1 Report

The re-submission of the article – even though it clarified several unclear aspects from its previous version – still misses the clarity and the self-contained style that would be required for a journal publication.

For examples, it is still unclear when each of the regression and classification losses are used.

The new version of Figure 1 is confusing once more: the notations are only to scare off the readership and their role as explaining the article is limited. For example, in the supplement the authors claim that the notation was clarified, yet there is no reference to the BN structure that appears in the image.

Similarly, in the description of equation 4, there are two factors, the weight and the \delta m_n that have the same role – therefore redundant.

The new Figure detailing the “deformable convolution” is not understandable in the sense that it does not relate to the explanatory equations (3) and (4).

The whole section 3.2 misses both the mathematical clarity and the understandability of the notions that are manipulated in the article.

Reviewer 4 Report

The authors have made an important effort to improve the paper, however, they have not considered some of my previous advice, and the paper still has some shortcomings:

1) The abstract uses the verb "used" for DCNs and spatial attention (line 4). However, some other parts of the paper, still use the verb "introduced" (i.e. as if these were novelties) for deformable convolutions and SA (for instance, lines: 66 and 145). I think this is not ONLY a matter of changing these two lines, but the authors should make it MUCH CLEARER what are their contributions, and what are improvements they TAKE from the literature to improve their version. So to really assess the novelty level of this paper.

I appreciate, however, the efforts made to explain the novelty of BAP-FPN, as well as the other citations added in lines 66-68, i.e. [14]-[16].

2) New experiments in Table 4 are performed with Faster-RCNN against the proposed AD-Faster-RCNN; however, Libra-RCNN and Cascade-RCNN are introduced in the literature review (and previous experiments) but not evaluated in these experiments. How does the proposed network perform against these newer RCNN variants?

Please consider calling these new experiments something else, instead of "additional experiment". Maybe, "Experiments on autonomous driving scenarios" or something similar. This should be the focus of the paper, and the text should also reflect this.

Also, please explain how were the 10k images taken from the BDD100k dataset. Is this their 10k subset? Or were this extracted by you from the test set of the 100k images? The protocol followed is not properly explained.

Round 3

Reviewer 4 Report

I think the paper is now fit for publication, congratulations.

Minor notes: please spell check the newly added paragraphs, as I detected some typos.